# Detection of Oxazolidinone Resistance Genes and Characterization of Genetic Environments in Enterococci of Swine Origin, Italy

**DOI:** 10.3390/microorganisms8122021

**Published:** 2020-12-17

**Authors:** Simona Fioriti, Gianluca Morroni, Sonia Nina Coccitto, Andrea Brenciani, Alberto Antonelli, Vincenzo Di Pilato, Ilaria Baccani, Simona Pollini, Lucilla Cucco, Alessandra Morelli, Marta Paniccià, Chiara Francesca Magistrali, Gian Maria Rossolini, Eleonora Giovanetti

**Affiliations:** 1Department of Biomedical Sciences and Public Health, Polytechnic University of Marche, 60121 Ancona, Italy; s.fioriti@pm.univpm.it (S.F.); g.morroni@univpm.it (G.M.); s.n.coccitto@pm.univpm.it (S.N.C.); 2Department of Experimental and Clinical Medicine, University of Florence, 50121 Florence, Italy; albertoanton88@gmail.com (A.A.); ila.baccani@gmail.com (I.B.); simona.pollini@unifi.it (S.P.); gianmaria.rossolini@unifi.it (G.M.R.); 3Clinical Microbiology and Virology Unit, Florence Careggi University Hospital, 50139 Florence, Italy; 4Department of Surgical Sciences and Integrated Diagnostics, University of Genoa, 16126 Genoa, Italy; vincenzo.dipilato@unige.it; 5Istituto Zooprofilattico Sperimentale dell’Umbria e delle Marche ‘Togo Rosati’, 06126 Perugia, Italy; l.cucco@izsum.it (L.C.); a.morelli@izsum.it (A.M.); m.paniccia@izsum.it (M.P.); c.magistrali@izsum.it (C.F.M.); 6Department of Life and Environmental Sciences, Polytechnic University of Marche, 60121 Ancona, Italy; e.giovanetti@univpm.it

**Keywords:** *Enterococcus faecium*, *Enterococcus faecalis*, *Enterococcus* spp., oxazolidinone resistance, *cfr*, *cfr*(D), *optrA*, *poxtA*, conjugative plasmid

## Abstract

One hundred forty-five florfenicol-resistant enterococci, isolated from swine fecal samples collected from 76 pig farms, were investigated for the presence of *optrA*, *cfr*, and *poxtA* genes by PCR. Thirty florfenicol-resistant *Enterococcus* isolates had at least one linezolid resistance gene. *optrA* was found to be the most widespread linezolid resistance gene (23/30), while *cfr* and *poxtA* were detected in 6/30 and 7/30 enterococcal isolates, respectively. WGS analysis also showed the presence of the *cfr*(D) gene in *Enterococcus faecalis* (*n* = 2 isolates) and in *Enterococcus avium* (*n* = 1 isolate). The linezolid resistance genes hybridized both on chromosome and plasmids ranging from ~25 to ~240 kb. Twelve isolates were able to transfer linezolid resistance genes to enterococci recipient. WGS analysis displayed a great variability of *optrA* genetic contexts identical or related to transposons (Tn*6628* and Tn*6674*), plasmids (pE035 and pWo27-9), and chromosomal regions. *cfr* environments showed identities with Tn*6644*-like transposon and a region from p12-2300 plasmid; *cfr*(D) genetic contexts were related to the corresponding region of the plasmid 4 of *Enterococcus faecium* E8014; *poxtA* was always found on Tn*6657*. Circular forms were obtained only for *optrA*- and *poxtA*-carrying genetic contexts. Clonality analysis revealed the presence of *E. faecalis* (ST16, ST27, ST476, and ST585) and *E. faecium* (ST21) clones previously isolated from humans. These results demonstrate a dissemination of linezolid resistance genes in enterococci of swine origin in Central Italy and confirm the spread of linezolid resistance in animal settings.

## 1. Introduction

Oxazolidinones, including linezolid and tedizolid, are antibiotics approved for clinical use to treat serious infections by Gram-positive pathogens, including MRSA, VRE, multidrug-resistant (MDR) pneumococci, and MDR mycobacteria. The oxazolidinones perturb the ribosomal peptidyl-transferase center and effect tRNA positioning resulting in inhibition of protein synthesis [1]. Therapeutic recommendations for oxazolidinone antibiotics include severe infections such as community and nosocomial pneumonia, bloodstream infections, and skin and soft tissue infections involving strains resistant to other drugs or in case of therapeutic failure [2].

Linezolid-resistant isolates were sporadically detected soon after the introduction of this antibiotic due to ribosomal mutations in 23S rRNA and/or in L3, L4, and L22 ribosomal proteins [3,4].

Moreover, transferable resistance mechanisms to oxazolidinones have emerged during the past decade. These mechanisms include: (*i*) post-transcriptional methylation of the 23S rRNA by the Cfr (chloramphenicol and florfenicol resistance) and Cfr-like methylases, which confer resistance to five classes of antimicrobial agents including phenicols, lincosamides, oxazolidinones, pleuromutilines and streptogramin A (PhLOPS_A_ phenotype) [5,6,7,8,9]; (*ii*) ribosomal protection by the ABC-F proteins OptrA (oxazolidinone phenicol transferable resistance) and PoxtA (phenicols, oxazolidinones and tetracyclines) leading to a decreased susceptibility to phenicols, oxazolidinones (including tedizolid), and tetracyclines (PoxtA only) [10,11,12].

In enterococci, of both human and animal origin, linezolid resistance genes are typically associated with mobile genetic elements, which facilitate their dissemination by horizontal transfer [10,13,14,15,16].

Florfenicol is widely used in veterinary medicine to treat infections in food-producing animals and as a result of co-selection events, could promote the spread not only of phenicol resistance genes but also of oxazolidinone resistance genes, with potentially serious consequences for human health [17].

Enterococci, natural inhabitants of the intestinal tract in humans and many animals, have been recognized as important hospital-acquired pathogens since many years [18]. *Enterococcus faecium* and *Enterococcus faecalis*, in particular, represent the second and third most important nosocomial pathogens worldwide [4]. Acquired resistance among enterococci and particularly resistance to glycopeptides is increasing, limiting the available therapeutic options. Oxazolidinones represent one of the few remaining treatment options for infections caused by VRE [4]. Besides the occurrence of linezolid-resistant enterococci in hospitals, their detection in other reservoirs is of special concern since the potential transfer of oxazolidinone resistance genes from enteric bacteria in animals to humans via the food chain could represent a serious public health problem.

Several reports on occurrence of transferable linezolid resistance genes in enterococci of animal origin are available, mostly from Asian countries [19,20,21,22,23,24], and only one from Italy [25], a country where the prevalence of VRE has been steadily increasing during the past five years [4].

Considering the higher use of antimicrobials in livestock in Italy compared to other European countries [26], the presence of oxazolidinones resistance genes and their genetic context in enterococci of swine origin in Italian Marche region was investigated, in order to gain insight into their dissemination dynamics.

## 2. Materials and Methods

### 2.1. Sampling Procedures and Enterococcal Isolation

Sampling was designed to be representative of pig production of the Marche region (central Italy), excluding small farms (Appendix A). Based on this criterion, a total of 76 swine herds, fattening farms (with at least 200 pigs) and breeding holdings (with at least 20 sows), were selected between April 2018 and July 2019.

Thirty-nine and 28 farms raised only finishers and weaners, respectively, while both pig categories were present in nine farms. Information about antibiotic use was obtained from farm registers, and was available for 60 out of 76 farms. In the year preceding sample collection, florfenicol, tetracycline, pleuromutilins, and lincomycin were administered on 27, 28, 16, and 34 farms, respectively. In each piggery, three pooled faecal samples were collected from finishers and/or weaners barns, in line with the protocols published by Hering et al. [27]. Briefly, on each farm, the barns housing finishers and/or weaners were divided in three groups, and one pooled sample of faeces was collected from each group. Overall, 255 samples were collected, 144 from finishers and 111 from weaners. The samples were stored at 4 °C before culturing, for a maximum of 8 h. Five grams of each homogenized sample were inoculated in 45 mL of buffered peptone water supplemented with florfenicol (10 mg/L) and incubated at 44 °C for 48 h. Then, 100 μL were inoculated on Slanetz-Bartley agar plates supplemented with florfenicol (10 mg/L). In case of growth, one enterococcal colony for each selective plate was randomly selected.

### 2.2. Genotypic and Phenotypic Characterization 

All selected florfenicol-resistant isolates were screened by PCR for the presence of *optrA*, *cfr*, and *poxtA* genes using primer pairs previously described [25]. The *cfr*-, *poxtA*-carrying *S. aureus* AOUC-0915 [28] and the *cfr*-, *optrA*-carrying *E. faecium* E35048 [29] isolates were used as positive controls in PCR experiments. The PCR products were subjected to Sanger sequencing.

Species identification of enterococci with at least one linezolid resistance gene was carried out by MALDI-TOF (Vitek-MS, bioMérieux, Marcy-l’Étoile, France) and confirmed by WGS data. Strains were tested for their susceptibility to florfenicol, chloramphenicol, linezolid, and tetracycline (SigmaAldrich, St. Louis, MI, USA) by standard broth microdilution assay, and to tedizolid using Etest strips (Liofilchem, Roseto degli Abruzzi, Italy). Susceptibility tests were interpreted according to EUCAST clinical breakpoints (EUCAST version 10.0, 2020) and CLSI breakpoint (CLSI M100-S27 document) [30]. *E. faecalis* ATCC 29212 was used as quality control.

### 2.3. S1-PFGE, Southern Blotting, and Hybridisation Assays

Total DNA embedded in agarose gel plugs, was digested with S1 nuclease (Thermo Fisher Scientific, Milan, Italy); chromosomal and plasmid DNAs were then separated by PFGE as previously described [31]. After S1-PFGE, total DNA was blotted onto positively charged nylon membranes (Ambion-Celbio, Milan, Italy) and hybridized with biotin-labelled probes specific for linezolid resistance genes as described elsewhere [32].

### 2.4. Detection of Circular Forms

To investigate the excision of the linezolid resistance genes contexts, PCR mapping assays were performed using outward-directed primer pairs targeting the linezolid resistance genes: (i) cfrdiv-FW acctgagatgtatggagaag and cfrdiv-RV gaatgagagagtagaaacgg; (ii) cfrD-FW ttcctaaaataaaacgacta and cfrD-RV tacaaaaagattcccagcca; (iii) optrAdiv-FW gaaaaataacacagtaaaaggc and optrAdiv-RV tttttccacatccatttctacc; (iv) poxtAdiv-FW gacgagccgaccaaccacct; and poxtAdiv-RV ttcaggcggacaaaaatccaa.

### 2.5. Conjugation Experiments

Conjugal transfer was performed on a membrane filter as described previously [32]. In mating experiments, all isolates carrying one or more linezolid resistance genes were used as donors. *E. faecium* 64/3, and *E. faecalis* JH2-2, two florfenicol-susceptible laboratory strains, were used as recipients [33]. Both the recipient and the donor were grown to an optical density of 0.4 ± 0.05 units at 540 nm and then mixed at a donor/recipient ratio of 1:5. The filter, placed on a pre-warmed plate of brain heart infusion agar (BHIA) (Oxoid, Basingstoke, UK), was incubated at 37 °C for 18 h. Cells were resuspended in 1 mL of sterile saline and plated onto BHIA supplemented with florfenicol (10 mg/L), fusidic acid (25 mg/L), and rifampicin (25 mg/L). Plates were incubated at 37 °C for 48 to 72 h and then examined for the presence of transconjugants. 

Transconjugants were tested for the presence of linezolid resistance genes by PCR and for their susceptibility to florfenicol, chloramphenicol, and linezolid. SmaI-PFGE was carried out and patterns analyzed to confirm the genetic background of transconjugants. Conjugation frequencies were expressed as number of transconjugants per recipient cell.

### 2.6. WGS and Sequence Analysis

All the isolates carrying at least one linezolid resistance gene were subjected to WGS analysis. Bacterial genomic DNA was extracted by the QIAcube automated extractor using DNeasy PowerLyzer PowerSoil Kit according to manufacturer’s instructions (Qiagen, Germany). Extracted DNA was subjected to WGS with an Illumina MiSeq platform (Illumina^®^, San Diego, CA, USA) using a 2 × 150 paired end approach. Raw reads were assembled using SPAdes-software [34]. In silico identification of acquired antimicrobial resistance genes, of ribosomal mutations involved in oxazolidinone resistance and clonal analysis were carried out using dedicated tools available at the Center for Genomic Epidemiology available at http://www.genomicepidemiology.org/ (e.g., MLST v.2.0, ResFinder v.3.2, LRE-finder v.1.0) and by the BLAST suite (https://blast.ncbi.nlm.nih.gov/Blast.cgi). eBURST was used to define clonal complexes (http://www.phyloviz.net/goeburst/).

Genetic relatedness among isolates was compared to all publicly available genomes of the same species and sequence type, for *E. faecalis* and *E. faecium* only, in the NCBI-NIH database and evaluated through SNP-based phylogenetic trees using Snippy v 4.4.3 with default parameters [35]. Maximum-likelihood (ML) phylogenetic trees were inferred from core SNP alignments by iq-tree v1.6.12 [36], using the K3P model with the ascertainment bias correction (ASC) for SNP data that typically do not contain constant sites. Branch supports were assessed by standard non-parametric bootstrap employing 100 replicate trees. The final ML tree with assigned support was visualized and edited using Microreact v48.0.0 (https://microreact.org/showcase).

### 2.7. Nucleotide Sequence Accession Numbers

The nucleotide sequences of *cfr* and *optrA* genetic contexts have been assigned to following GenBank accession numbers: MT723949 to MT723965, MW013838, and MW013839.

## 3. Results and Discussion

### 3.1. Detection of Oxazolidinone Resistance genes in Florfenicol-Resistant Enterococci and Antimicrobial Susceptibility Profiles

One hundred forty-five out of a total of 255 (56.9%, CI 95%: 50.6–63%) collected faecal pools, were positive for the presence of florfenicol-resistant enterococci, with no difference between the prevalence among finishers (79/144, 64.9%) and weaners (64/111, 57.7%).

PCR screening showed that 30 of 145 isolates, from 23 farms including 15 *E. faecalis*, seven *E. faecium*, four *Enterococcus avium*, two *Enterococcus hirae*, one *Enterococcus gallinarum*, and one *Enterococcus thailandicus*, carried at least one linezolid resistance gene (Table 1). 

Overall, *optrA* was the most prevalent linezolid resistance gene, being detected in 23/30 enterococci. The analysis of the deduced OptrA sequences revealed that four isolates harbored the wild-type OptrAE349, identical to that from *E. faecalis* E349 [10], while nineteen carried eight OptrA variants, five of which were original (Table 2).

*cfr* and *poxtA* genes were found in 6/30 and 7/30 isolates, respectively. The *cfr* gene that was not detected in *E. faecium*, was usually associated to the *optrA* gene (5/6) (Table 1); however, its relevance in *Enterococcus* spp. has not been clearly defined [39]. It should be noted that the *optrA* gene was closely associated with the *E. faecalis* species as it was found in 15/15 isolates. This data confirmed a role of *E. faecalis* to serve as a reservoir for spreading *optrA* in the human setting, as underlined by Deshpande et al. [39]. Furthermore, among the 15 enterococci belonging to species other than *E. faecalis*, *optrA* and *poxtA* genes were found in 8/15 (*E. faecium*, *E. avium*, *E. gallinarum,* and *E. hirae*) and in 7/15 (*E. faecium* and *E. hirae*) isolates, respectively (Table 1).

The *poxtA* gene, first described in an MRSA of clinical origin and subsequently in a porcine *E. faecium* isolated in Italy [11,25], has been recently reported in 19 enterococci from swine and chicken and in 66 porcine enterococci in China [23,40], and in six enterococci from retail meat and food-producing animals from Tunisia [41]. The presence of *poxtA* in enterococci of animal origin, mainly *E. faecium* species [23,41], suggested that selection of this gene could have occurred in the animal setting owing to the extensive use of phenicols in veterinary medicine [19].

No mutations involving the 23S rRNA or ribosomal proteins were detected either on assembled genomes or using LRE-finder software.

The thirty enterococci showed either susceptibility or resistance to linezolid (MIC range, 0.5–8 mg/L) and tedizolid (MIC range, 0.25–4 mg/L), and resistance to florfenicol (MIC range, 16–>128 mg/L), chloramphenicol (MIC range, 16–>128 mg/L), and tetracycline (MIC range, 32–>128 mg/L) (Table 1).

WGS analysis highlighted also the presence of the *cfr*(D) gene in *E. faecalis* S221 and S377 isolates and in *E. avium* S193 (Table 1). It was identical to the gene reference sequence in the plasmid 4 of *E. faecium* E8014 (GenBank accession no. LR135354.1) [42]. The *cfr*(D) gene has already been found in a clinical *E. faecalis* isolate in Spain [43], while to the best of our knowledge this is the first detection of *cfr*(D) in *E. avium* of swine origin. The *cfr*(B), *cfr*(C), and *cfr*(E) genes were not detected.

### 3.2. Relatedness of Isolates

The MLST analysis for *E. faecalis* identified eight known STs (ST16, ST27, ST73, ST108, ST314, ST330, ST476, and ST585); of which four ST16, ST27, ST476, and ST585 have been associated with human enterococci [19,39,41,44,45,46]. 

WGS analysis showed that *E. faecalis* isolates of the same ST were closely related to each other (e.g., ST27, ST108, ST73, ST314, and ST476, SNP range 2–62) (Appendix A). In two cases, related strains were collected from the same farm (e.g., strains belonging in ST27, ST108). In other cases (e.g., strains belonging in ST314, ST73, and ST476), strains were collected from different farms that, only for ST73, were of the same company (Table 1).

Regarding strains relatedness with already sequenced enterococci, the *E. faecalis* isolates were overall highly related (SNP range 36–107) with strains of animal origin (e.g., strains belonging in ST108, ST314, and ST476) from different countries (including USA, Brazil, and Tunisia), while the ST16 strain was more closely related to strains from different sources (i.e., of environmental and human origin) (Appendix A). 

Detection in swine of *E. faecalis* of human origin raises worrisome questions about possible strain transmission between animals and humans. 

*E. faecium* isolates exhibited seven distinct lineages based on MLST analysis: ST21, ST142, ST184, ST957, ST1534, and ST1667, all largely distributed in an animal setting, and a novel ST1734 [19,43]. The eBURST analysis revealed that ST21 belonged to the clonal complex CC17, one of the most widespread *E. faecium* nosocomial clone. Two other clonal complexes were detected: the first one included ST957 and ST1667 (dual locus variant) while the second comprised ST184 and ST1534 (dual locus variant). A core-genome SNP-based analysis performed with the ST21 *E. faecium* showed that the most similar was a human isolate from USA (SNPs *n* = 182), while the others (*n* = 8) were less closely related (SNPs *n* = 530–3786) (Appendix A).

Interestingly, the two *E. hirae* isolated from two different farms but derived from the same company were almost identical (SNPs *n* = 1) and both showed a high degree of similarity with a Canadian isolate of animal origin (SNPs *n* = 7–8). On the contrary, the four *E. avium* collected from different farms showed a higher diversity (SNPs *n* = 17901–25698), while S252 showed similarities with two isolates from Germany and one from Mongolia (SNPs *n* = 226–232). The single strain of *E. thailandicus* retrieved from AP3-PV farm was not related to the other representatives of the same specie (SNPs *n* = 7574–9638). Finally, the *E. gallinarum* strain showed a high degree of similarity with isolates from different geographical areas, mostly of animal origin (SNPs *n* = 17–79) (Appendix A).

### 3.3. Location of Oxazolidinones Resistance Genes

*optrA* was located on chromosome (*n* = 7 isolates), on plasmids (*n* = 9 isolates), or on both (*n* = 7 isolates); the plasmids ranged from 25 to 240 kb in size (Table 3). In the *optrA*-carrying enterococci, where hybridization occurred on both chromosome and plasmids, the gene was likely located on mobile genetic elements able to move intracellularly. Since their movement could involve an excision event resulting in a circular intermediate, we screened for the presence of circular intermediates by inverse PCR. *optrA*-carrying minicircles were detected in 17 of 23 isolates. These results could suggest an intracellular mobility of the *optrA*-carrying elements due to the IS-mediated recombination.

In five of six enterococci, the *cfr* gene was located on plasmids of ~97 kb (*n* = 1 isolate), ~48 kb (*n* = 3 isolates), or ~23 kb (*n* = 1 isolate) in size; in one strain, *cfr* showed a chromosomal location (Table 3). No circular intermediate was detected.

*cfr*(D) was detected on plasmids of ~30 and ~80 kb (*n* = 1 isolate), and ~34 kb (*n* = 2 isolates). In all strains, the *cfr*(D) gene also had a chromosomal location (Table 3). No circular form was detected.

In all but one *poxtA*-carrying enterococci, a plasmidic localization of this gene was detected; 5/7 isolates showed a double hybridization reaction on plasmids of different sizes [~30 kb and ~97 kb, and ~30 kb and ~150 kb]. In two isolates, *poxtA* hybridized on a ~97 kb plasmid and on the chromosome, respectively (Table 3). In all the isolates circular forms were detected.

Notably, four strains showed co-localisation of linezolid resistance determinants: i.e., *optrA* and *cfr* were located in a ~48 kb plasmid in S176, and in a ~97 kb plasmid in S251. In S157, *optrA* and *poxtA* were located on the same plasmids of ~30 and ~97 kb. Finally, in S221 *optrA* and *cfr*(D) were co-located on a plasmid of ~80 kb.

### 3.4. Transferability of Oxazolidinones Resistance Genes

Seven out of 23 *optrA*-carrying isolates successfully transferred *optrA* gene alone (*n* = 4 isolates) or in association with *cfr* (*n* = 1), with *cfr*(D) (*n* = 1), or with *poxtA* (*n* = 1) in intra- and interspecific mating experiments (Table 3).

Five of seven *poxtA-*positive enterococci were able to transfer this linezolid resistance gene alone (*n* = 4) or in association with *optrA* (*n* = 1, see above) to *E. faecalis* JH2-2 or *E. faecium* 64/3 recipients. The isolates belonging to *E. gallinarum* (*n* = 1) and *E. thailandicus* (*n* = 1) species were unable to transfer linezolid resistance genes to enterococcal recipients at detectable frequencies.

MICs and genotypes for both transconjugants and donors, and transfer frequencies are indicated in Table 3.

Contigs of different sizes obtained by WGS from the 30 enterococci with at least one oxazolidinone resistance gene were analyzed. Genetic background of *optrA*, *cfr*, *cfr*(D), and *poxtA* genes are briefly described below and shown in Figures 1–4. The information concerning the gene localization is reported in Table 3.

#### 3.4.1. Genetic Contexts of the *optr*A Genes

The *optrA* genetic contexts showed high variability and could be categorized into six different types, designated A (*n* = 11 isolates), B (*n* = 5 isolates), C (*n* = 2 isolates), D (*n* = 1 isolate), E (*n* = 2 isolates), and F (*n* = 2 isolates) (Figure 1A–F). 

**Type A:** The *optrA* genetic context detected in S377 was identical to Tn*6628* transposon, first identified in an *E. faecium* clinical isolate integrated into the pE35048-oc plasmid (Figure 1A) [15].

Seven enterococci harbored a Tn*6628-*like transposon where one of the two IS*Efa15*, the one upstream of *optrA* in S176, or that downstream of *optrA* in S183, S184, S217, S219, S221, and S345, was truncated.

Tn*6628* transposon was entirely rearranged in S340, where in fact the two IS*Efa15* were inverted compared to the wild-type transposon and in the one downstream of *optrA* a transposase was deleted.

In S193, the *optrA*-carrying region, not flanked by the IS*Efa15* elements, instead of being integrated into the gene encoding for the ζ toxin protein as in pE35048-oc plasmid, was located between *erm*(B) and an intact gene for the ζ toxin. 

Even in S155, the *optrA* context was not bracketed by IS*Efa15*; the region downstream of *optrA* gene showed high identity with the first six ORFs of the pE35048-oc plasmid.

**Type B:** Four enterococci (S224, S232, S338, and S339) showed a chromosomal *optrA*-carrying context inserted into the *radC* gene and exhibiting a high degree of identity (99%) to multiresistance transposon Tn*6674* first identified in the chromosome of a porcine *E. faecalis* (accession no. MK737778) (Figure 1B) [47].

A Tn*6674*-like transposon was detected in S325; the transposon likewise inserted into the *radC* gene was smaller due to the loss of *fexA-*carrying region (2475 bp) replaced by a DNA fragment (1439 bp) containing a hypothetical protein (Figure 1B). Notably, to the best of our knowledge, this is the first time a Tn*6674*-like transposon is detected in a species other than *E. faecalis.*

**Type C:** In S297, the *optrA* genetic environment (13,168 bp in size), displayed a high degree of identity (99%) to the chromosomal “*optrA* gene cluster” region from *E. faecalis* LY4 (accession no. KT862785.1) (Figure 1C) [44]. The region between *optrA* and *erm*(A)-like genes was found also in *E. faecalis* S341; however, in this isolate, the *optrA* environment, including also the *fexA* gene, was bracketed by two IS*1216* in the opposite orientation (Figure 1C).

**Type D:** In S157, the *optrA* gene, close to IS*1542* element, was found in a large region (13,226 bp) exhibiting a high degree of identity to two chromosomal area of *Staphylococcus sciuri* wo33-13 containing *optrA* and Tn*558*, respectively (accession no. KX982174.1) [48]. However, in S157, these two regions were reversed compared to those in *S. sciuri* wo33-13 chromosome (Figure 1D).

**Type E:** This *optrA* genetic context was observed in S251 and S252 isolates. The *optrA* environment (16,588 bp in size), flanked by two IS*1216*, showed three areas of identity with the pWo27-9 plasmid of *S. sciuri* (accession no. KX982169) [48]. In both isolates, the *optrA* gene, showing opposite orientation than that in pWo27-9, was linked to *cfr* (bracketed by *istA*-*istB* element and a transposase), *ble,* and *aph(2″)-IIIa* resistance genes (Figure 1E). 

**Type F:** A 12,345-bp *optrA* genetic context, bracketed by two IS*1216* in the opposite orientation, was detected in S380 and S381 isolates. Two regions were found to exhibit a high degree of identity (99%) to the corresponding of pE035 plasmid of *E. faecalis* E035 (accession no. MK140641.1) (Figure 1F) [40].

#### 3.4.2. Genetic Contexts of the *cfr* Genes

The *cfr* genetic context detected in S176 exhibited a high degree of identity (99%) to the Tn*6644* region including one copy of IS*Enfa5* and *cfr* gene; this transposon in *Staphylococcus aureus* AOUC-0915 is part of Tn*6349*, in turn inserted in the chromosome (accession no. MH746818.1) (Figure 2) [16].

In S325, the region including the *cfr* gene exhibited a high degree of identity (99%) to different areas: (*i*) the *fexA*-containing region (6.5 kb) of *S. epidermidis* p12-02300 plasmid (accession no. KM521837.1) [49], and (*ii*) the *cfr-*carrying region of Tn*6644* of *S. aureus* AOUC-0915; however, upstream of the *cfr* gene a single IS*Enfa5* was detected, which was shorter due to a 396-bp deletion in the first transposase (Figure 2).

In S173 the *cfr* environment (10,990 bp), bracketed by two IS*1216* in the same orientation, was similar to that of p12-2300; however, *cfr* was located upstream *tnpB* and a 1.6-kb insertion was identified between *cfr* and *fexA* genes (Figure 2).

The *cfr* genetic context (8462 bp) in S155 was flanked by two IS*S1N* in the same orientation. *cfr* and *fexA* genes, compared to those in p12-2300, showed opposite orientation (Figure 2).

The *cfr* environments of S251 and S252 isolates, co-carrying *optrA* gene, are described above and shown in Figure 1E.

#### 3.4.3. Genetic Contexts of the *cfr*(D) Genes

The *cfr*(D) context detected in *E. faecalis* S221 and *E. hirae* S193 was identical to the corresponding region in the plasmid 4 of *E. faecium* E8014 except for the IS element upstream of *guaA* (IS*S1N* instead of IS*1216*) [42]. In *E. faecalis* S337, the *cfr*(D) genetic context was flanked by ΔIS*S1E* and IS*S1A* elements and showed the presence of a truncated *guaA* gene downstream of *cfr*(D) (Figure 3).

#### 3.4.4. Genetic Context of the *poxt*A Gene

The *poxtA* context detected in seven isolates was always flanked by two IS*1216* and identical to the corresponding region of Tn*6657* transposon from *S. aureus* AOUC-0915 (Figure 4) [16].

## 4. Conclusions

Although oxazolidinones are not approved for veterinary use, linezolid resistance genes have been detected in animal and environmental bacteria worldwide, likely due to their co-selection by different agents (e.g., phenicols) that are also affected by the encoded resistance determinants (e.g., OptrA, Cfr, and PoxtA proteins). The present study, to the best of our knowledge the first carried out in Italy, provided data concerning the occurrence of linezolid resistance genes in enterococci of swine origin, suggesting their possible role as a reservoir for major human pathogens. Plasmids and other mobile genetic elements apparently play a key role in the spread of oxazolidinone resistance genes among swine enterococci, as suggested by the remarkable diversity of bacterial lineages and genetic contexts to which the main oxazolidinone resistance genes were found to be associated.

In fact, it is now apparent that humans, animals, and the environment are interconnected, and the emergence of clinically-relevant resistance determinants in humans may be the tip of the iceberg of a more widespread phenomenon.

To control the emergence and spread of oxazolidinones resistance in the clinical setting, therefore, a “One Health” approach is called for. An effective control of oxazolidinones resistance needs integrated approach to better know the evolution of linezolid-resistant enterococci and their genetic determinants, the transmission routes, and mechanisms involved.

## Figures and Tables

**Figure 1 microorganisms-08-02021-f001:**
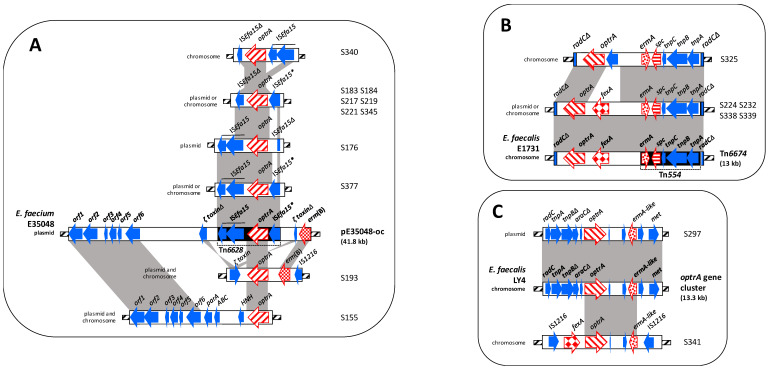
Schematic representation (not to scale) of the *optrA* genetic environment of 23 *optrA*-carrying enterococci compared to known genetic elements (shown in bold): pE35048-oc from *E. faecium* E35048, (**A**); Tn*6674* from *S. aureus* AOUC-0915, (**B**); chromosomal *optrA* gene cluster from *E. faecalis* LY4, (**C**); *S. sciuri* Wo33-13 chromosome, (**D**); pWo27-9 from *S. sciuri* Wo27-9, (**E**); pE035 from *E. faecalis* E035, (**F**). Arrows indicate the positions and directions of transcription of the different genes. Antibiotic resistance genes are displayed in different red textures. Grey-shaded areas represent regions of >99% nucleotide sequence identity. In each box, the *optrA* localisation (on the left) and the isolates (on the right) were shown.

**Figure 2 microorganisms-08-02021-f002:**
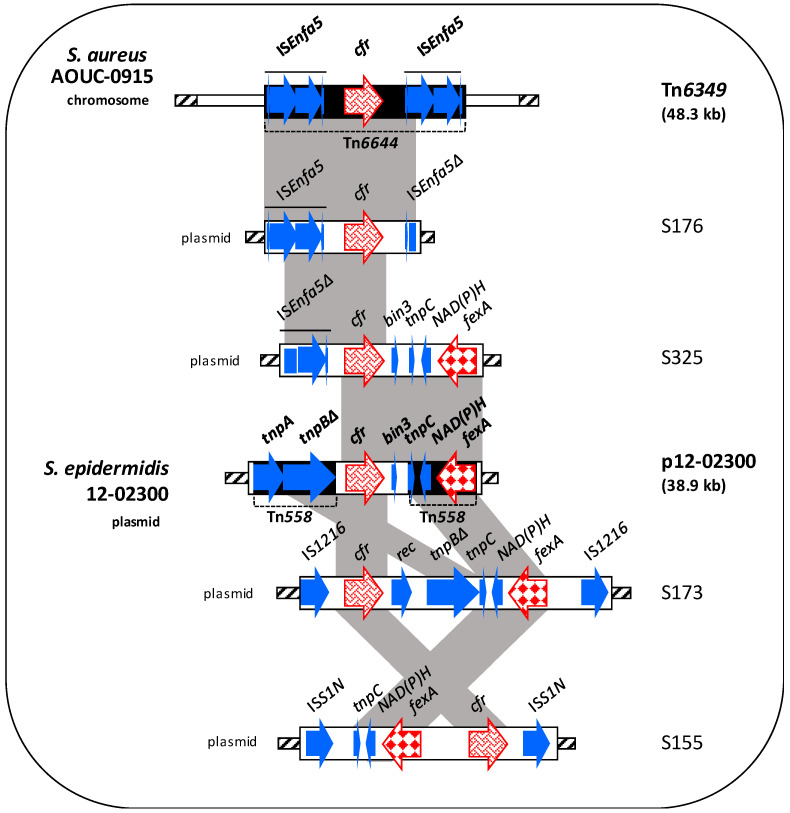
Schematic representation (not to scale) of the *cfr* genetic environment of 4/6 *cfr*-carrying enterococci compared to known genetic elements (shown in bold). Arrows indicate the positions and directions of transcription of the different genes. Antibiotic resistance genes are displayed in different red textures. Grey-shaded areas represent regions of >99% nucleotide sequence identity. In the box, the *cfr* localisation (on the left) and the isolates (on the right) were shown. The *cfr* genetic contexts of S251 and S252 isolates are shown in Figure 1E.

**Figure 3 microorganisms-08-02021-f003:**
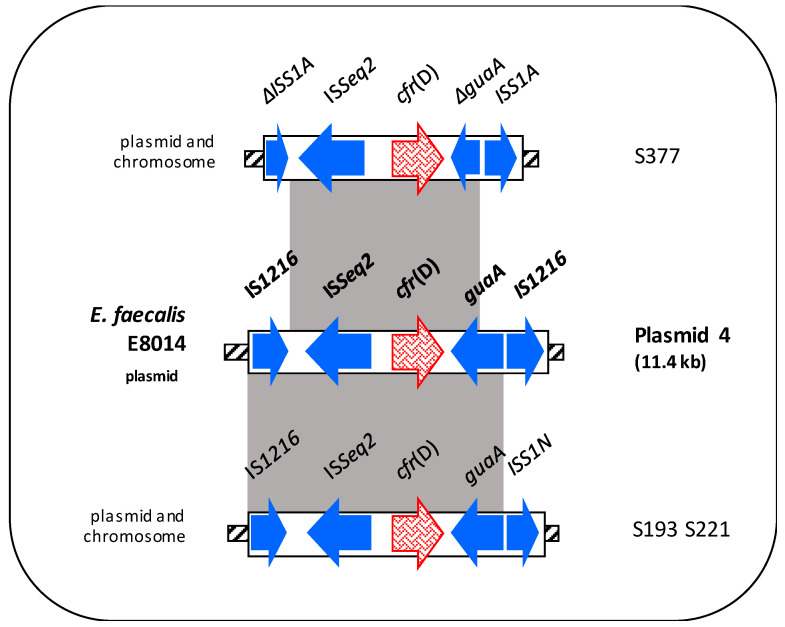
Schematic representation (not to scale) of the *cfr*(D) genetic environment of the 3 *cfr*(D)-carrying enterococci compared to a region of plasmid 4 of *E. faecalis* E8014 (shown in bold). Arrows indicate the positions and directions of transcription of the different genes. Antibiotic resistance genes are displayed in different red textures. Grey-shaded areas represent regions of >99% nucleotide sequence identity. In the box, the *cfr*(D) localization (on the left) and the isolates (on the right) were shown.

**Figure 4 microorganisms-08-02021-f004:**
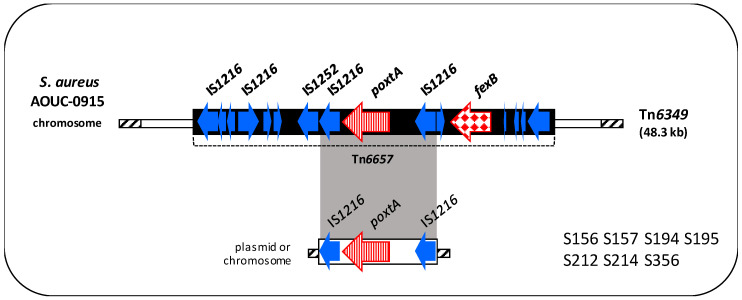
Schematic representation (not to scale) of the *poxtA* genetic environment of the 7 *poxtA*-carrying enterococci compared to Tn*6657* transposon (shown in bold). Arrows indicate the positions and directions of transcription of the different genes. Antibiotic resistance genes are displayed in different red textures. Grey-shaded areas represent regions of >99% nucleotide sequence identity. In the box, the *poxtA* localisation (on the left) and the isolates (on the right) were shown.

**Table 1 microorganisms-08-02021-t001:** Linezolid resistance genes, antimicrobial susceptibility profiles, and typing data of 30 enterococci.

Strain	Species	Oxazolidinone Resistance Genes				MIC (mg/L)					MLST	Farm	Area	Company
		*optrA*	*cfr*	*cfrD*	*poxtA*	FFC ^a^	CHL	LZD	TZD	TE				
S183	*E. faecalis*	+	-	-	-	32	64	2	1	64	ST27	AP13 ^b^-W ^c^	4	D
S184	*E. faecalis*	+	-	-	-	64	128	2	2	64	ST27	AP13-W	4	D
S217	*E. faecalis*	+	-	-	-	32	64	2	2	128	ST330	AP27-W	4	0
S219	*E. faecalis*	+	-	-	-	32	128	4	1.5	128	ST314	AN1-F ^d^	2	E
S221	*E. faecalis*	+	-	+	-	32	64	2	2	64	ST314	AN1-W	2	E
S224	*E. faecalis*	+	-	-	-	64	128	4	2	128	ST476	AN2-W	2	E
S232	*E. faecalis*	+	-	-	-	128	128	2	3	128	ST476	AN5-F	2	0
S251	*E. faecalis*	+	+	-	-	64	16	4	2	64	ST16	PS2-F	1	0
S338	*E. faecalis*	+	-	-	-	64	64	1	2	128	ST476	MC9-F	3	C
S339	*E. faecalis*	+	-	-	-	128	>128	4	4	128	ST476	MC9-F	3	C
S340	*E. faecalis*	+	-	-	-	128	>128	4	2	128	ST314	MC12-W	3	C
S341	*E. faecalis*	+	-	-	-	>128	>128	8	4	128	ST585	MC12-W	3	C
S377	*E. faecalis*	+	-	+	-	32	32	4	2	128	ST73	MC17-W	3	0
S380	*E. faecalis*	+	-	-	-	64	128	4	2	>128	ST108	MC20-W	3	0
S381	*E. faecalis*	+	-	-	-	128	128	2	2	>128	ST108	MC20-W	3	D
S156	*E. faecium*	-	-	-	+	>128	32	8	4	128	ST1667	MC1-W	3	0
S157	*E. faecium*	+	-	-	+	>128	32	8	3	32	ST1534	AP1-F	5	A
S194	*E. faecium*	-	-	-	+	64	16	4	1.5	128	ST142	AP18-F	4	A
S195	*E. faecium*	-	-	-	+	64	16	2	1	128	ST184	AP19-F	5	D
S297	*E. faecium*	+	-	-	-	>128	64	4	1.5	128	ST957	AP30-F	5	0
S345	*E. faecium*	+	-	-	-	128	>128	4	2	128	ST21	MC8-F	3	A
S356	*E. faecium*	-	-	-	+	>128	64	4	1.5	128	ST1734	MC13-W	3	C
S212	*E. hirae*	-	-	-	+	128	64	8	1.5	128	-	AP24-F	5	A
S214	*E. hirae*	-	-	-	+	128	32	8	1.5	128	-	AP25-F	4	A
S173	*E. avium*	-	+	-	-	32	32	1	0.25	32	-	AP12-F	4	C
S176	*E. avium*	+	+	-	-	32	16	2	1.5	32	-	AP7-F	4	0
S193	*E. avium*	+	-	+	-	16	16	1	0.5	32	-	AP18-F	4	D
S252	*E. avium*	+	+	-	-	32	16	4	0.75	32	-	PS2-F	1	0
S325	*E. gallinarum*	+	+	-	-	64	64	0.5	0.75	>128	-	MC2-W	3	E
S155	*E. thailandicus*	+	+	-	-	>128	>128	4	2	64	-	AP3-W	4	A

^a^ FFC, florfenicol; CHL, chloramphenicol; LZD, linezolid; TZD, tedizolid; TE, tetracycline. ^b^ Farm code. ^c^ W, Weaners. ^d^ F, Finishers. +: presence of the gene. -: absence of the gene.

**Table 2 microorganisms-08-02021-t002:** OptrA variants detected in 23 *Enterococcus* isolates.

OptrA Sequence		Strain	ST	References
Variant	Amino Acid Substitutions			
Wild type (OptrA_E349_)	-	*E. faecalis* S224 *E. faecalis* S338 *E. faecalis* S339 *E. faecalis* S232	ST476 ST476 ST476 ST476	[10]
DP	Y176D T481P	*E. faecalis* S341	ST585	[37]
KD	T112K Y176D	*E. faecalis* S380 *E. faecalis* S381	ST108 ST108	[37]
DD	Y176D G393D	*E. faecium* S297	ST957	[37]
DDM	Y176D G393D I622M	*E. faecium* S157 *E. gallinarum* S325	ST1534 -	This study
EYD	K3E D12Y Y176D	*E. faecalis* S251 *E. avium* S252	ST16 -	This study
EYKWDVDASKELYNKQLEIG	K3E N12Y N122K Y135W Y176D A350V G393D V395A A396S Q509K Q541E M551L N560Y K562N Q565K E614Q I627L D633E N640I R650G	*E. thailandicus* S155 *E. avium* S176 *E. faecium* S345	- - ST21	[38]
EYKWKVDASKELYNKQLEIG	K3E N12Y N122K Y135W I287K A350V G393D V395A A396S Q509K Q541E M551L N560Y K562N Q565K E614Q I627L D633E N640I R650G	*E. faecalis* S183 *E. faecalis* S184 *E. faecalis* S217 *E. faecalis* S219 *E. faecalis* S221 *E. faecalis* S340	ST27 ST27 ST330 ST314 ST314 ST314	This study
EYKCKDVDASKELYNKQLEIG	K3E N12Y N122K Y135C E161K Y176D A350V G393D V395A A396S Q509K, Q541E, M551L, N560Y, K562N, Q565K, E614Q, I627L, D633E, N640I, R650G	*E. avium* S193	-	This study
YKCKDSVDASKELYNKQLEIG	K3E N12Y N122K Y135W Y176D A350V G393D V395A A396S Q509K Q541E M551L N560Y K562N Q565K E614Q I627L D633E N640I R650G	*E. faecalis* S377	ST73	This study

**Table 3 microorganisms-08-02021-t003:** Florfenicol, chloramphenicol, and linezolid MICs and resistance genotypes for enterococci donors and relevant transconjugants. Localization of linezolid resistance genes and detection of circular forms.

Donor/ Transconjugant	Recipient	Frequency	MICs (mg/L)			Genotype				Gene Localization (Detection Circular Form)			
			FFC ^a^	CHL	LZD	*optrA*	*cfr*	*cfrD*	*poxtA*	*optrA*	*cfr*	*cfrD*	*poxtA*
***E. faecalis* S183**	*E. faecium* 64/3	ND ^b^	32	64	2	+	-	-	-	100 ^c^ (+) ^d^	-	-	-
	*E. faecalis* JH2-2	1.1 × 10^−2^											
TC1			32	64	2	+	-	-	-				
TC2			32	64	2	+	-	-	-				
***E. faecalis* S184**	*E. faecalis* JH2-2	1.6 × 10^−2^	64	128	2	+	-	-	-	100(+)	-	-	-
TC3			32	128	4	+	-	-	-				
TC6			32	128	4	+	-	-	-				
***E. faecalis* S217**	*E. faecalis* JH2-2	ND	32	64	2	+	-	-	-	90(+)	-	-	-
***E. faecalis* S219**	*E. faecalis* JH2-2	ND	32	128	4	+	-	-	-	90(+)	-	-	-
***E. faecalis* S221**	*E. faecalis* JH2-2	ND	32	64	2	+	-	+	-	80(+)	-	c, 30, 80(-)	-
***E. faecalis* S224**	*E. faecalis* JH2-2	ND	64	128	4	+	-	-	-	c ^e^ (+)	-	-	-
***E. faecalis* S232**	*E. faecalis* JH2-2	ND	128	128	2	+	-	-	-	c, 90(+)	-	-	-
***E. faecalis* S338**	*E. faecalis* JH2-2	ND	64	64	1	+	-	-	-	c(-)	-	-	-
***E. faecalis* S339**	*E. faecalis* JH2-2	ND	128	>128	4	+	-	-	-	c(+)	-	-	-
***E. faecalis* S340**	*E. faecalis* JH2-2	ND	128	>128	4	+	-	-	-	c(+)	-	-	-
***E. faecalis* S341**	*E. faecalis* JH2-2	ND	>128	>128	8	+	-	-	-	c(-)	-	-	-
***E. faecalis* S377**	*E. faecalis* JH2-2	ND	32	32	4	+	-	+	-	c, 40(+)	-	c, 34(-)	-
***E. faecalis* S380**	*E. faecalis* JH2-2	ND	64	128	4	+	-	-	-	70(-)	-	-	-
***E. faecalis* S381**	*E. faecalis* JH2-2	ND	128	128	2	+	-	-	-	c, 70, 90(-)	-	-	-
***E. faecalis* S251**	*E. faecalis* JH2-2	ND	64	16	4	+	+	-	-	97(+)	97(-)	-	-
***E. faecium* S156**	*E. faecium* 64/3	3.5 × 10^−4^	>128	32	8	-	-	-	+	-	-	-	30, 97(+)
1A1			128	32	8	-	-	-	+				
1A2			128	32	8	-	-	-	+				
***E. faecium* S157**	*E. faecium* 64/3	4.6 × 10^−4^	>128	32	8	+	-	-	+	c, 30, 50, 97(+)	-	-	30, 97(+)
1B1			128	32	8	+	-	-	+				
1B2			128	32	8	+	-	-	+				
***E. faecium* S194**	*E. faecium* 64/3	3.7 × 10^−4^	64	16	4	-	-	-	+	-	-	-	30, 97(+)
TC21			64	16	4	-	-	-	+				
TC22			64	16	4	-	-	-	+				
***E. faecium* S195**	*E. faecium* 64/3	7.3 × 10^−2^	64	16	2	-	-	-	+	-	-	-	30, 150(+)
TC6			64	16	4	-	-	-	+				
TC8			64	16	4	-	-	-	+				
***E. faecium* S297**	*E. faecalis* JH2-2	ND	>128	64	4	+	-	-	-	240(+)	-	-	-
	*E. faecium* 64/3	1.8 × 10^−4^											
TC36			64	64	2	+	-	-	-				
TC37			64	64	2	+	-	-	-				
***E. faecium* S345**	*E. faecium* 64/3	1.8 × 10^−3^	128	>128	4	+	-	-	-	c, 45, 150(+)	-	-	-
TC25			128	64	4	+	-	-	-				
TC27			128	64	4	+	-	-	-				
***E. faecium* S356**	*E. faecalis* JH2-2	ND	>128	64	4	-	-	-	+	-	-	-	c(+)
***E. hirae* S212**	*E. faecalis* JH2-2	ND	128	64	8	-	-	-	+	-	-	-	97(+)
***E. hirae* S214**	*E. faecium* 64/3	1.3 × 10^−3^	128	32	8	-	-	-	+	-	-	-	30, 97(+)
TC9			64	32	8	-	-	-	+				
TC12			64	32	8	-	-	-	+				
	*E. faecalis* JH2-2	ND											
***E. avium* S173**	*E. faecalis* JH2-2	6.1 × 10^−3^	32	32	1	-	+	-	-	-	48(-)	-	-
TC1			16	32	1	-	+	-	-				
TC2			16	32	1	-	+	-	-				
***E. avium* S176**	*E. faecium* 64/3	7 × 10^−5^	32	16	2	+	+	-	-	48(+)	48(-)	-	-
1C1			64	16	4	+	+	-	-				
	*E. faecalis* JH2-2	1.5 × 10^−3^											
TC1			64	16	4	+	+	-	-				
TC2			64	16	4	+	+	-	-				
***E. avium* S193**	*E. faecalis* JH2-2	1.1 × 10^−4^	16	16	1	+	-	+	-	c, 48(-)	-	c, 34(-)	-
TC4			16	32	0.5	+	-	+	-				
TC5			16	32	0.5	+	-	+	-				
***E. avium* S252**	*E. faecalis* JH2-2	ND	32	16	4	+	+	-	-	c(-)	c(-)	-	-
***E. gallinarum* S325**	*E. faecalis* JH2-2	ND	64	64	0.5	+	+	-	-	c(+)	23(-)	-	-
***E. thailandicus*S155**	*E. faecium* 64/3	ND	>128	>128	4	+	+	-	-	c, 25, 97(+)	48(-)	-	-

^a^ FFC, florfenicol; CHL, chloramphenicol; LZD, linezolid; TZD, tedizolid; TE, tetracycline. ^b^ ND, not detectable transfer under conditions used. ^c^ plasmid size (in kb). ^d^ detected (+) or not detected (-) circular form. ^e^ c, chromosome. 3.4. Genetic environments of oxazolidinone resistance genes.

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
