# Peer review of "Detection of Oxazolidinone Resistance Genes and Characterization of Genetic Environments in Enterococci of Swine Origin, Italy"

_microorganisms, 2020, doi:10.3390/microorganisms8122021_

Round 1

Reviewer 1 Report

The manuscript by Morroni et al. (Manuscript ID: microorganisms-987942) is well written, logically organized and provides information on the ecology of oxazolidinone resistant enterococci of swine origin, Italy. However, this manuscript requires a modification/ revision before accepting for publication.

Comments:

L85: Please provide details of the amount of fecal sample (size in grams) used to inoculate in buffered peptone in this study. Also, provide details of the sampling procedure, transport, storage, etc.

A35: Provide brief information on the profile of antibiotics used on 76 hog farms in order to strengthen the logical case of this study.

L97: Explain the rationale for using antimicrobials in this study?

L98: Please provide a valid reference to microdilution method used in this study (e.g., CLSI M100-S27, 2017).

L118: The statement is confusing to readers: E. faecium 64/3 and E. faecalis JH2-2, both were originally isolated from human stool samples. Please refer Nicolas et al. 2006 (FEMS Microbiol Lett. 254, 27-33). Also, describe filter mating assay.

L168: PoxtA gene was found in E. faecium and E. hirae isolates and not widespread? Please clarify/ update.
L227: the plasmids ranged from 25 to 240 kb in size (table 3) doesn’t match with Table 3?  Please update. Similarly, L234, L237 and 241 doesn’t match with Table 3.

L247: This should be “thirteen out of 23 optrA-carrying isolates”

L258: Table 3: wrong foot note. Also, add footnote “e”.

Figure 4: this should be Tn6657 not 6349.

L47&50 and elsewhere: When used for the first time, please describe the gene name in parenthesis. E.g., poxtA (a novel phenicol-tetracycline resistance gene).

Author Response

Reviewer 1

The manuscript by Morroni et al. (Manuscript ID: microorganisms-987942) is well written, logically organized and provides information on the ecology of oxazolidinone resistant enterococci of swine origin, Italy. However, this manuscript requires a modification/ revision before accepting for publication.

Response: We thank the reviewer for the positive comments. All issues were addressed in the revised manuscript.

Comments:

L85: Please provide details of the amount of fecal sample (size in grams) used to inoculate in buffered peptone in this study. Also, provide details of the sampling procedure, transport, storage, etc.

Response: We edited the section including the information about sample collection and processing. Thank you for the opportunity to clarify this point (lines 89-94).

A35: Provide brief information on the profile of antibiotics used on 76 hog farms in order to strengthen the logical case of this study.

Response: We agree with the reviewer that data on antibiotic consumption would have provided interesting information in this paper, therefore we decide to include the number of farms using the main antibiotic classes relevant for the scope of this study in the year before sample collection. It should be noted that 16 out of 76 farmers did not agree to provide access to the paper registers, therefore data on antibiotic consumption was reported for 60 farms (lines 85-88).

L97: Explain the rationale for using antimicrobials in this study?

Response: The acquired resistance genes cfr, optrA and poxtA confer resistance to phenicols and oxazolidinones (cfr only for linezolid, while optrA and poxtA for linezolid and tedizolid). The poxtA gene also causes reduced sensitivity to tetracyclines. Therefore, all acquired linezolid resistance genes also confer resistance to phenicols. Even if oxazolidinones are for human use only, phenicols might still promote to spread of linezolid resistant determinants among animal bacteria.

L98: Please provide a valid reference to microdilution method used in this study (e.g., CLSI M100-S27, 2017).

Response: A reference to microdilution method has been added to the revised manuscript (line 107).

L118: The statement is confusing to readers: E. faecium 64/3 and E. faecalis JH2-2, both were originally isolated from human stool samples. Please refer Nicolas et al. 2006 (FEMS Microbiol Lett. 254, 27-33). Also, describe filter mating assay.

Response: The suggested reference and a brief description of the filter mating assay have been added to the revised manuscript (lines 126-131).

L168: PoxtA gene was found in E. faecium and E. hirae isolates and not widespread? Please clarify/ update.

The manuscript has been modified accordingly (lines 178-180).

L227: the plasmids ranged from 25 to 240 kb in size (table 3) doesn’t match with Table 3?  Please update.

Response: we reviewed both manuscript and table 3, but the size range for optrA plasmids is correct: from 25 kb (in E. thailandicus S155) to 240 kb (in E. faecium S297).

Similarly, L234, L237 and 241 doesn’t match with Table 3.

Response: we reviewed both manuscript and table 3, the data are correct.

L247: This should be “thirteen out of 23 optrA-carrying isolates”

Response: We apologize for the mistake the correct number is “seven out of 23 optrA-carrying isolates…” (line 257).

L258: Table 3: wrong foot note. Also, add footnote “e”.

Response: The reviewer is right we apologize for the mistake. The foot note of table 3 has been modified accordingly.

Figure 4: this should be Tn6657 not 6349.

Response: Tn6657, that is part of Tn6349, is shown (highlighted in with a dashed parenthesis) in Figure 4. Please see reference # 16.

L47&50 and elsewhere: When used for the first time, please describe the gene name in parenthesis. E.g., poxtA (a novel phenicol-tetracycline resistance gene).

Response: The manuscript has been modified as suggested by reviewer (lines 49, 52-53).

Reviewer 2 Report

The authors undertook to demonstrate the oxazilidinone resistance genes among Enterococcus strains isolated from pigs and to explain the genetic background of this resistance. The manuscript has been prepared at a good level, the subject matter is very important in the aspect of the „One Health” concept and so far poorly recognized resistance to this group of antimicrobials among strains isolated from animals.

I only suggest minor corrections:

The title is too general, it should be noted that the manuscript is not only about the detection of oxazolidinone resistance genes

Line 71: I suggest to support the sentence with a reference

Line 86 and 88: it should be explained why the authors used the concentration of 10 mg / L and what media and temperature controls were used (the possibility of growth inhibition on Slanetz Bartley with an antibiotic addition, as well as higher incubation temperature could inhibit the growth of some Enterococcus species)

Line 150-151: Is this a subheading?

Line 191: please spell "8"

Author Response

Reviewer 2

The authors undertook to demonstrate the oxazilidinone resistance genes among Enterococcus strains isolated from pigs and to explain the genetic background of this resistance. The manuscript has been prepared at a good level, the subject matter is very important in the aspect of the „One Health” concept and so far poorly recognized resistance to this group of antimicrobials among strains isolated from animals.

Response: We thank the reviewer for the positive comments. All issues were addressed in the revised manuscript.

I only suggest minor corrections:

The title is too general, it should be noted that the manuscript is not only about the detection of oxazolidinone resistance genes

Response: The title has been modified as suggested by reviewer (lines 2-3).

Line 71: I suggest to support the sentence with a reference

Response: The reference #4 has been added to the revised manuscript (line 73).

Line 86 and 88: it should be explained why the authors used the concentration of 10 mg / L

Response: There are no florfenicol breakpoints for Enterococcus (antibiotic approved exclusively for animal use), however in all papers the florfenicol-resistant strains selection is carried out with 10 mg/L. Please see: Liu et al. Vet Microbiology 2014 (doi:10.1016/j.vetmic.2014.02.037) and Li et al. JAC 2019 (doi:10.1093/jac/dkz449).

and what media and temperature controls were used (the possibility of growth inhibition on Slanetz Bartley with an antibiotic addition, as well as higher incubation temperature could inhibit the growth of some Enterococcus species)

We used the same conditions of a previous work aimed to isolate enterococci from stool samples. Please see: Del Grosso et al. MDR 2000 (doi:10.1089/mdr.2000.6.313).

Line 150-151: Is this a subheading?

Response: The reviewer is right. It is an editorial mistake (Lines 160-161).

Line 191: please spell "8"

Response: The number “8” has been modified accordingly (line 203).

Reviewer 3 Report

The manuscript describes analysis of antibiotic resistance genes spread in farm animals in Italy by a combination of methods. The analysis is performed properly and the data are presented clearly.

In terms of novelty, the manuscript does not add anything to basic science. It is a typical microbiology paper reporting results of a routine screen. There is no discussion or results except a summary of results. Can authors include a short discussion of results?

Author Response

Reviewer 3

The manuscript describes analysis of antibiotic resistance genes spread in farm animals in Italy by a combination of methods. The analysis is performed properly and the data are presented clearly.

In terms of novelty, the manuscript does not add anything to basic science. It is a typical microbiology paper reporting results of a routine screen.

Response: We thank the reviewer for the comments. The issue was addressed in the revised manuscript. 

There is no discussion or results except a summary of results. Can authors include a short discussion of results?

Discussion of the various results was already provided within the related paragraphs in the “Results and Discussion” section. However, we expanded the conclusions section according to the suggestion of the Reviewer (lines 371-374, 376-382 and 383-386).